# FEW-SHOT LEARNING WITH SIMPLEX

## ABSTRACT

Deep learning has made remarkable achievement in many fields. However, learning the parameters of neural networks usually demands a large amount of labeled data. The algorithms of deep learning, therefore, encounter difficulties when applied to supervised learning where only little data are available. This specific task is called few-shot learning. To address it, we propose a novel algorithm for few-shot learning using discrete geometry, in the sense that the samples in a class are modeled as a reduced simplex. The volume of the simplex is used for the measurement of class scatter. During testing, combined with the test sample and the points in the class, a new simplex is formed. Then the similarity between the test sample and the class can be quantized with the ratio of volumes of the new simplex to the original class simplex. Moreover, we present an approach to constructing simplices using local regions of feature maps yielded by convolutional neural networks. Experiments on Omniglot and miniImageNet verify the effectiveness of our simplex algorithm on few-shot learning.

## 1 INTRODUCTION

Deep learning has exhibited outstanding ability in various disciplines including computer vision, natural language processing and speech recognition (LeCun et al., 2015). For instance, AlexNet has made a breakthrough on recognizing millions of imagery objects by means of deep Convolutional Neural Network (CNN) (Krizhevsky et al., 2012). In the past five years, the algorithmic capability of comprehending visual concepts has been significantly improved by elaborately well-designed deep learning architectures (He et al., 2015a; Szegedy et al., 2014). However, training deep neural networks such as the widely employed CNNs of AlexNet (Krizhevsky et al., 2012), Inception (Szegedy et al., 2015), VGG (Simonyan & Zisserman, 2014), and ResNet (He et al., 2015b), needs the supervision of many class labels which are handcrafted. For example, the number of samples of each class in the ImageNet of object recognition benchmark (Russakovsky et al., 2015) is more than one thousand. In fact, the number of labelled samples used for learning parameters of CNNs is far more than that because data augmentation is usually applied. This kind of learning obviously deviates from the manner of human cognition. A child can recognize a new object that she/he has never seen only by several examples, from simple shapes like rectangles to highly semantic animals like tigers. However, deep learning algorithms encounter difficulty in such scenarios where only very sparse data are available for learning to recognize a new category, thus raising the research topic of one-shot learning or few-shot learning (Fei-Fei et al., 2003; Tenenbaum et al., 2011).

The seminal work Fei-Fei et al. (2006) models few-shot learning with the Bayesian framework. Empirical knowledge of available categories is learned and parameterized as a probability density function. The unseen class with a handful of examples is modeled as the posterior by updating the prior. Bayesian theory provides a simple and elegant idea for solving learning problems with little data. If decomposed into parts or programs, an object can be described by the joint distribution of Bayesian criterion. In this manner, human-level performance on one-shot learning has been derived for discovering simple visual concepts such as ancient handwritten characters (Lake et al., 2015).

With the prevalence of deep learning, the recent work for few-shot learning focuses on the application of deep neural networks that have more capacity to accommodate the complexity of object representations. Siamese neural network facilitates the performance of few-shot recognition by means of twin networks of sharing parameters, optimizing the distances of representative features in intra-classes (Koch et al., 2015). The counterpart of learning data structures by distance is also formulated by triplet loss in Lin et al. (2017). Researchers in Lin et al. (2017) assert that the distance metrics

can learn the intrinsic manifold structures of training data such that the network is more general and robust when employed for untrained objects. A very recent work pertaining to distance-based optimization, named Prototypical Networks (Snell et al., 2017), significantly improves the capability of few-shot recognition. Prototypical Networks attempt to minimize the distance of the test sample to the center of each class and are learned in the end-to-end manner.

Memory-augmented architectures are also proposed to help assimilate new classes with more accurate inference (Santoro et al., 2016). Matching network embeds metric learning in neural network in the light of attention mechanism which is embodied by softmax (Vinyals et al., 2016). In a very recent work, the large-scale memory without the need of resetting during training is formulated as an embedded module for arbitrary neural networks to remember the information of rare events (Kaiser et al., 2017). In order to obtain rapid learning with limited samples, meta learning is exploited both in memory network and matching network. This "learning to learn" technique is extended to deal with few-shot learning from the point of view of optimization (Ravi & Larochelle, 2017). To be specific, a LSTM-based meta learner learns to mimic the exact optimization algorithm and then harnesses the acquired capability to train the learner applied for the few-shot cases. The latest meta learning algorithms also deal with few-shot learning from different angles, e.g. the fast adaptation of neural networks (Finn et al., 2017), and temporal convolution (Mishra et al., 2017).

In addition to the application of memory module or attention model in LSTM, there is another type of algorithms digging the effective way of transferring the discriminative power of pre-trained models to few-shot circumstances. Resorting to the correlation between the activations in the last feature layers and the associated parameters for softmax, a transformation is learned to derive the parameters for predicting new classes from corresponding activations (Qiao et al., 2017).

The algorithms based on deep learning can learn more expressive representations for objects, essentially boosting the quality of feature extraction. However, the softmax classifier discriminates all categories by class boundaries, bypassing the steps that carefully characterize the structure of each class. Thus the algorithmic performance will deteriorate grossly if the distribution of new class cannot be accurately modeled by trained networks. Besides softmax, another commonly applied method, $k$ nearest neighbors (KNN), is a point-to-point measurement and is incapable of conveying global structural information.

To address this issue, we propose a geometric method for few-shot learning. Our perspective is that accurate geometric characterization for each class is essential when only a handful of samples are available, because such sparse data are usually insufficient to fit well-converged parameterized classifier. To this end, we harness convex polytope to fit a class, in the sense that we construct a convex polytope by selecting the samples in the class as the vertices of the polytope. The volume of the polytope is taken as the measurement of class scatter. Thus the polytopal volume may be improved after including the query sample in the test set during the testing trial. The normalized volume with respect to the original counterpart is applied to compute the distance from the test sample to the test set. To highlight the structural details of object parts, we present the construction of polytope based on convolutional feature maps as well.

To the best of our understanding, however, there is no exact formula to calculating the volume of general convex polytope. To make our algorithm feasible, therefore, we use the simplest convex polytope — *simplex* instead. The volume of a simplex can be expressed by the Cayley-Menger determinant (Cayley, 2009), thus casting the problem of few-shot recognition as a simple calculation of linear algebra. Experiments on Omniglot and miniImageNet datasets verify the effectiveness of our simple algorithm.

## 2 MODELING CLASS WITH SIMPLEX

It is known that by proper embedding, the feature representations of images or documents in the same class spatially cluster together. Each feature vector with a length of $d$, in its nature, corresponds to a point in $d$-dimensional space. We model each class as a polytope, with feature representation of each data point in this class as its vertex.

Our approach is based on the idea that feature vectors of the test sample will be close to the polytope of its own class, formed by feature vectors of the corresponding examples, and distant from the

others. Hence, we can perform the classification by finding the 'nearest' class polytope to which the test point belongs, using carefully designed distance metrics associated with the content of polytope.

As we point out in the introduction, there is no exact solution to computing the volume of a polytope. Therefore, we resort to the simplex to accomplish our idea. The simplex is the simplest counterpart of convex polytope and its volume admits a closed form expression. So we focus our attention on simplex to develop our algorithm.

## 2.1 SIMPLEX

A simplex is the conceptual extension of a triangle in high-dimensional spaces. To be formal, let $\hat{\mathcal{Y}} = \{y_0, y_1, \ldots, y_n\}$ denote a set of points in $\mathbb{R}^d$. A simplex is the convex polytope with the condition of $n = d$, implying that there needs exact $d + 1$ points to constitute a simplex in the $d$-dimensional space. For convenience, we call such a simplex the $d$-simplex. For instance, a line is a 1-simplex, a triangle is a 2-simplex, and a tetrahedron is a 3-simplex. Moreover, a line has the length, a triangle has the area, and a tetrahedron has the volume. In convention, we use the contents to represent the length, the area, and the volume (Weisstein, 2002).

A particularly intriguing property of the simplex is that its content can be written in a closed form by virtue of the Cayley-Menger determinant (Cayley, 2009). To show this, let $A = [y_1 - y_0, \ldots, y_n - y_0]$ and the Cayley-Menger matrix

$$\hat{P} = \begin{bmatrix} 0 & e^T \\ e & P \end{bmatrix}, \tag{1}$$

where $e$ denotes the all-one column vector of length $n + 1$, $T$ presents the transpose of a matrix or a vector, and the entry $P_{ij}$ of the distance matrix $P$ is of form $P_{ij} = \|y_i - y_j\|^2$. The content of simplex $\hat{\mathcal{Y}}$ has two expressions that coincide with each other, showing that

$$C^2(\hat{\mathcal{Y}}) = \frac{1}{(n!)^2} \det(A^T A) = \frac{-1}{(-2)^n (n!)^2} \det(\hat{P}), \tag{2}$$

where $\det(A^T A)$ is the Gram determinant and $\det(\hat{P})$ is the Cayley-Menger determinant. Our analysis is based on the application of formula (2).

## 2.2 DISTANCE MEASUREMENT WITH SIMPLEX

Let $\mathcal{Y} = \{y_1, \ldots, y_n\}$ be the feature set of an arbitrary class. These features can be derived from outputs of deep neural networks, e.g., CNN. It is clear that $\hat{Y} = \{\mathcal{Y} \cup y_0\}$. Let $t$ denote a test sample. It is clear the content $C(\mathcal{Y})$ of the corresponding simplex[1] will be large if data points in $\mathcal{Y}$ are sparse and small if compact. Therefore, $C(\mathcal{Y})$ is a plausible measurement for the class scatter.

An exact $d$-simplex will be formed during testing process if the test sample is merged into $\mathcal{Y}$. Then the associated content will be improved from $C(\mathcal{Y})$ to $C(\mathcal{Y} \cup t)$. The incremental content will be marginal if the feature point of the test sample is close to the class simplex, meaning the high correlation of the spatial proximity. Then the dissimilarity measurement of one test sample to one class can be written as

$$\left(\ell(t, \mathcal{Y})\right)^{\frac{1}{2}} = \frac{C(\mathcal{Y} \cup t)}{C(\mathcal{Y})}. \tag{3}$$

Here the numerator $C(\mathcal{Y})$ serves to eliminating the quantitative influence of the class scatter. The normalization is indispensable because for a large $C(\mathcal{Y})$, the incremental content $C(\mathcal{Y} \cup t) - C(\mathcal{Y})$ will be prone to be relatively large even if the test sample is close to the simplex.

To make it clear, we explicitly write $\ell(t, \mathcal{Y})$. Let the Cayley-Menger matrix pertaining to simplex $\mathcal{Y}$ be

$$\hat{Q} = \begin{bmatrix} 0 & e^T \\ e & Q \end{bmatrix}, \tag{4}$$

---

[1] In fact, it is not a $d$-simplex here because the number of data points is equal to the dimension. But the reduced simplex still has the content. We will explain the theory in section 2.3.

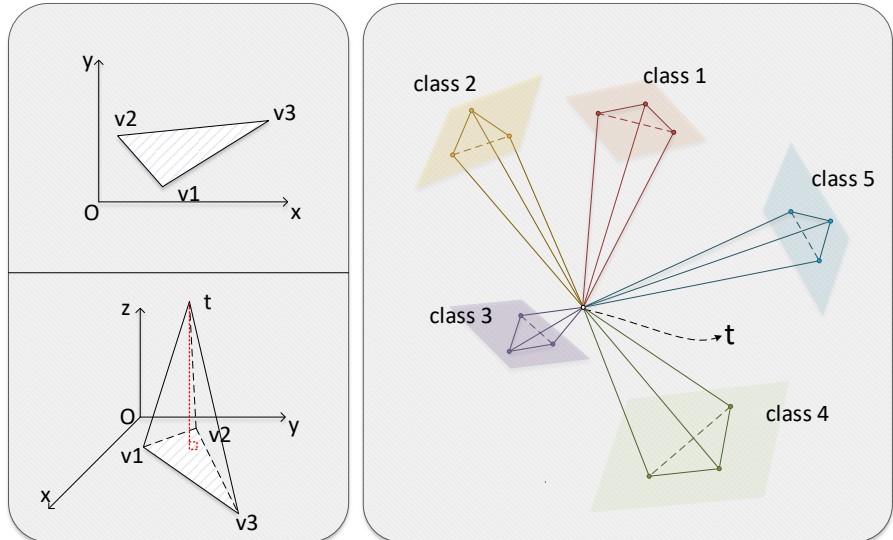

Figure 1: Visualization of simplices formed during a test episode in a k-way n-shot classification task, where $k=5$ and $n=3$. The points with the same color correspond to the same category and cluster together. Within each class, $k$ vertices form a $(k-1)$-simplex (a triangle in the 2-dimensional plane). With the same embedding function, the test sample point $t$ is also mapped to this space. Combined with one class, a new simplex (tetrahedron in the 3-dimensional space) is obtained, located in the space of one dimension higher than the original one. It is not surprising that the test sample point $t$ is close to its own class while distant from the others, representing a certain degree of intra-class similarities and inter-class diversities.

where $Q_{ij} = \|y_i - y_j\|^2$ and $i, j = 1, \ldots, n$. Then the content of the simplex formed by $\mathcal{Y}$ can be written as

$$C^2(\mathcal{Y}) = \frac{-1}{(-2)^{n-1}((n-1)!)^2} \det(\hat{Q}). \tag{5}$$

Substituting (2) and (5) into (3), we derive

$$\ell(t, \mathcal{Y}) = -\frac{1}{2n^2} \frac{\det(\hat{P})}{\det(\hat{Q})}, \tag{6}$$

where $y_0 = t$ is implicitly assumed for $\hat{P}$. It is straightforward to know that the smaller $\ell(t, \mathcal{Y})$ is, the closer the test point to the class simplex.

To help intuitively understand our algorithm, a visual schematic example is shown in Figure 1, where the complete procedure is figuratively demonstrated.

## 2.3 ISOMETRIC EMBEDDING OF SIMPLEX

It is obvious that $\mathcal{Y}$ cannot form a $d$-simplex due to the number $|\mathcal{Y}|$ of data points in $\mathcal{Y}$ satisfies $|\mathcal{Y}| = d$, violating the definition of the $d$-simplex. However, our model for the few-shot learning can proceed without any modification. To make this clear, we need to introduce the isometric embedding of the simplex.

The points on a geometric entity $\mathbb{R}^{d_e}$ (manifolds or polytopes) can be expressed with coordinates when the entity is placed in an ambient space $\mathbb{R}^{d_a}$. Usually, the intrinsic dimension $d_e$ is much less than the ambient dimension $d_a$, especially when $d_a$ is large. Formally, there exists a function

$$f : \mathcal{R}^{d_e} \to \mathbb{R}^{d_a}$$
$$\tau \to f(\tau)$$

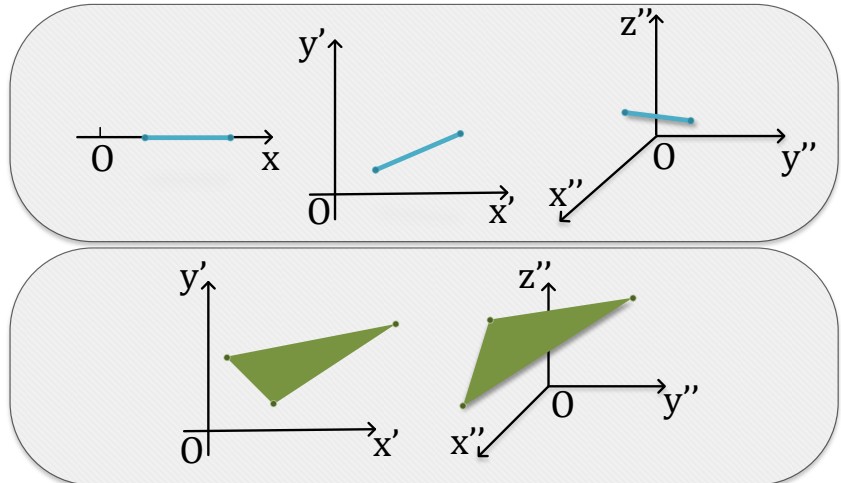

Figure 2: Isometric embedding preserves the geometric properties of a simplex.

For the question that we are interested in, $\mathcal{R}^{d_e}$ is a simplex. Both $\mathcal{R}^{d_e}$ and $\mathbb{R}^{d_a}$ are Euclidean. The isometric embedding means $\|\tau_i - \tau_j\| = \|f(\tau_i) - f(\tau_j)\|$ for an arbitrary pair of vertices. For a simplex $\mathcal{S}$ in $\mathcal{R}^{d_e}$, therefore, the contents of $C(\mathcal{S})$ and $C(f(\mathcal{S}))$ satisfy the identity $C(\mathcal{S}) = C(f(\mathcal{S}))$ under isometric embedding. That is to say, we can correlate the dimension of the simplex with the number of vertices under isometric embedding in the case where the number of vertices is insufficient to construct a $d_a$-simplex. This reduced simplex in the $\mathbb{R}^{d_a}$ is obvious a low-dimensional one.

Two visual instances are illustrated in Figure 2, where a segment line of fixed length is a 1-simplex. It can be embedded in the 1-dimensional, 2-dimensional and 3-dimensional spaces, respectively. Certainly, we can apply the coordinates of different dimensions for these two endpoints. However, its length is constant in different ambient spaces due to the isometric embedding. When a simplex is mapped into a higher dimensional space by an isometric embedding, the coordinates of its vertices vary, whereas its geometric properties pertaining to distance remain unchanged. Therefore we can employ formula (6) to perform our algorithm.

For few-shot classification tasks, such reduced simplices are ubiquitous because feature representations of data points embedded by variously parameterized functions are always of high dimension while the number of available examples for a class is quite few. For example, images classification tasks generally require deep convolutional networks (such as VGGs or Inception) as the embedding function. The VGG16 network produces a long vector of $4096$ dimensions after the final fully-connected layer. For a five-shot learning, however, there are only five examples in each class. Hence, the reduced simplices we actually exploit are of dimension four for the computation of $C(\mathcal{Y})$ and dimension five for that of $C(\mathcal{Y} \cup t)$.

The above geometric analysis is reminiscent of the theoretic assumption in manifold learning (Tenenbaum et al., 2000; Roweis & Saul, 2000), especially the Isomap algorithm (Tenenbaum et al., 2000). Interested readers may refer to these two seminal papers in manifold learning for further understanding.

## 2.4 METRIC ANALYSIS

It is known that matching network is the extension of metric learning via attention model. Here we analyze the metric characteristic of our algorithm. This angle of analysis may be useful for digging deep insights of few-shot learning. To this end, we need to reveal the details of formula (6), which is described in Theorem 1.

**Theorem 1.** *The geometric dissimilarity measurement $\ell(t, \mathcal{Y})$ from point $t$ to $\mathcal{Y}$ can be expanded as the following form*

$$\ell(t, \mathcal{Y}) = -\frac{1}{2n^2} \frac{e^T P^{-1} e}{e^T Q^{-1} e} p^T Q^{-1} p, \tag{7}$$

*where $p_i = \|y_i - t\|^2$ and*

$$P = \begin{bmatrix} 0 & p^T \\ p & Q \end{bmatrix}. \tag{8}$$

*Proof.* For matrices $S_{11}$, $S_{12}$, $S_{21}$, and $S_{22}$, the Schur's determinant identity is the form

$$\det \begin{bmatrix} S_{11} & S_{12} \\ S_{21} & S_{22} \end{bmatrix} = \det(S_{22}) \det(S_{11} - S_{12}(S_{22})^{-1} S_{21}). \tag{9}$$

Applying the Schur's determinant identity twice, we can obtain the expanded form of $\det(\hat{P})$

$$\det(\hat{P}) = -e^T P^{-1} e \det(P) = e^T P^{-1} e p^T Q^{-1} p \det(Q). \tag{10}$$

By the same way, we can also obtain the analogous form of $\det(\hat{Q})$

$$\det(\hat{Q}) = -e^T Q^{-1} e \det(Q). \tag{11}$$

Substituting equations (10) and (11) into equation (6), we achieve the expansion of the measurement $\ell(t, \mathcal{Y})$ in (7). This concludes the proof of Theorem 1. $\square$

It is readily to know that both $Q$ and $P$ are not positive definite matrices. Hence $Q^{-1}$ and $P^{-1}$ cannot be viewed as metric matrices in (7) because there are negative eigenvalues. However, there still exists the explicit algebraic meaning. The first factor $w_1 = e^T P^{-1} e / e^T Q^{-1} e$ is an incremental ratio with respect to the class and the second factor $w_2 = p^T Q^{-1} p$ is equivalent to the summation of a kind of scale-normalized distance transformation.

## 3    Feature Representation by local regions

Motivated by the success of applying the decomposed parts in (Lake et al., 2015), we present an approach to constructing a simplex using the spatial feature maps of CNNs. Moreover, the convolutional feature maps have been effectively employed for visual relational reasoning in (Santoro et al., 2017).

While applying deep convolutional networks in classification tasks, the tensor (feature maps) will be flatten to be a 1-D feature vector in the last convolution layer and then it is fed into the fully connected layers. This process makes the feature maps lose adjacency information of spatial structures in the 2-D image.

Although flattening seems inevitable in conventional classification pipeline, for few-shot tasks, it is necessary to collect information more effectively from multiple local regions for making an accurate classification decision. In order to play down the negative effect of flattening operation, besides the flattened feature vectors, we take fully advantage of feature maps by applying various scales of attention windows for the last convolution layer. Within each local region, a $3 \times 3$ region on $5 \times 5$ feature maps for instance, we perform the same flattening operation, generating a feature vector containing local information. Using this strategy, we are able to generate more feature vectors for simplex construction with few samples. Figure 3 clearly displays the operation of sampling feature vectors on feature maps with local regions.

To further carry out this idea of preserving the adjacency information, we tag the $l^2$ $d$-dimensional cells with their corresponding coordinates indicating their relative spatial locations. With the extra 2 dimensions, we are able to restrain these cells on an underlying manifold, retaining their relative positions as in the original image. With such a simple manipulation, our model shows an appreciable improvement in performance.

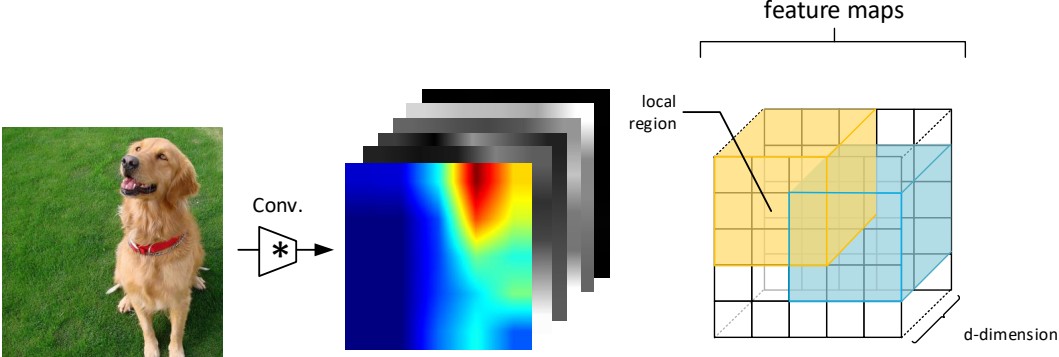

Figure 3: Illustration of sampling feature vectors from feature maps in the last convolution layer. The feature maps corresponding to the different local regions are flattened to be 1-D feature vectors as the vertices of the simplex for the associated class.

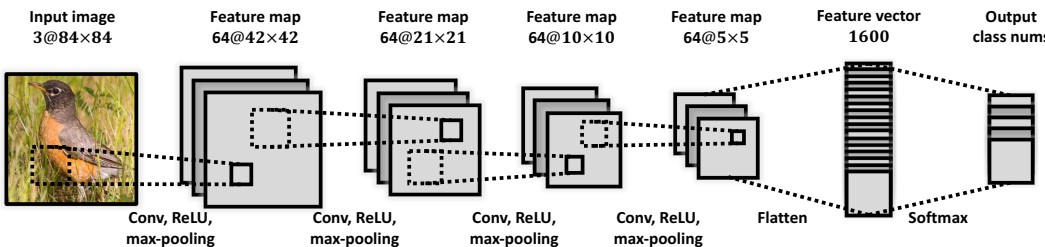

Figure 4: The architecture of the 4-block CNN used in our experiments.

## 4 EXPERIMENTS

In regard to the experiment setup, the number of examples in each class should be constrained so as to fit the "few-shot" scenario. A typical experiment setting is the $N$-way $k$-shot task (Vinyals et al., 2016): for each of the $N$ new categories, $k$ examples are provided. Given a set of unlabeled test samples, the model needs to classify them into these $N$ categories. Since the number of available examples is limited (e.g. 1 or 5), training deep convolutional networks either from the scratch or with fine-tuning on new class data will generally lead to over-fitting. We performed our model on two different datasets: miniImageNet (Vinyals et al., 2016) and Omniglot (Lake et al., 2015).

The algorithms to be compared fall into two categories: conventional methods and deep learning methods. For deep learning methods, we compare our simplex algorithm with three state-of-the-art ones: matching networks (Vinyals et al., 2016), Meta-Learner LSTM (Ravi & Larochelle, 2017), and prototypical networks (Snell et al., 2017). Essentially, our algorithm is to measure the distance between a data point and a data set. For conventional methods, therefore, we take the Mahalanobis distance (Mahalanobis, 1936) and Minimum Incremental Coding Length (MICL) (Wright et al., 2008) for comparison. The MICL algorithm can be used to measure the structural similarity by coding theory of multivariate Gaussian data.

A simple four-block CNN is employed to learn the representations of data both for miniImageNet and Omniglot. The architecture follows the learner network proposed by Ravi & Larochelle (2017), which contains four $3 \times 3$ convolutional layers with 64 filters. Each is followed by batch normalization, ReLU activation and $2 \times 2$ max-pooling. Following all the above layers is one fully connected layer, and lastly a softmax layer with the number equal to the number of classes being trained upon. The output is optimized with a cross-entropy loss function by the Adam optimizer with a learning rate of 0.001. The architecture is shown in Figure 4. The traditional algorithms compared are all performed on features extracted by this four-layer CNN.

Table 1: Few-shot validation of different local regions on miniImageNet (95% confidence interval).

| Size of local regions | 5-way 1-shot Acc. | 5-way 5-shot Acc. |
|:---:|:---:|:---:|
| $1 \times 1$ | $40.88\% \pm 0.44\%$ | – |
| $2 \times 2$ | $\mathbf{42.66\% \pm 0.48\%}$ | $58.21\% \pm 0.47\%$ |
| $3 \times 3$ | $42.34\% \pm 0.42\%$ | $\mathbf{58.98\% \pm 0.40\%}$ |
| $4 \times 4$ | $40.76\% \pm 0.43\%$ | $58.02\% \pm 0.44\%$ |
| $5 \times 5$ | $32.85\% \pm 0.70\%$ | $58.63\% \pm 0.68\%$ |

Table 2: Few-shot accuracy on miniImageNet on $95\%$ confidence interval. *Reported by Ravi & Larochelle (2017)

| Model | Fine Tune | 5-way 1-shot Acc. | 5-way 5-shot Acc. |
|:---|:---:|:---:|:---:|
| Baseline KNN | N | $32.69\% \pm 0.75\%$ | $40.32\% \pm 0.68\%$ |
| Mahalanobis Distance | N | $32.39\% \pm 0.70\%$ | $60.09\% \pm 0.39\%$ |
| MICL (Wright et al., 2008) | N | $42.59\% \pm 0.46\%$ | $61.06\% \pm 0.36\%$ |
| | | | |
| Matching networks* | N | $43.40\% \pm 0.78\%$ | $51.09\% \pm 0.71\%$ |
| Matching networks FCE* | N | $43.56\% \pm 0.84\%$ | $55.31\% \pm 0.73\%$ |
| Meta-Learner LSTM* | N | $43.44\% \pm 0.77\%$ | $60.60\% \pm 0.71\%$ |
| Prototypical networks (Snell et al., 2017) | N | $49.42\% \pm 0.78\%$ | $68.20\% \pm 0.66\%$ |
| | | | |
| **Simplex (ours)** | N | $46.93\% \pm 0.43\%$ | $62.00\% \pm 0.26\%$ |

## 4.1 MINIIMAGENET

ImageNet is a large-scale image database designed for multiple computer vision tasks. Since it would be extremely time-consuming to test the few-shot performance on the full ImageNet, we turn to miniImageNet instead (Vinyals et al., 2016), the subset of ImageNet with 100 categories selected. For each category, 600 images with size $84 \times 84$ are provided. Following the same split as Ravi & Larochelle (2017), miniImageNet is divided into a 64-class training set, a 16-class validation set and a 20-class test set. We train the embedding network on the 64-class training set and validate the result based on the 16-class validation set. The 20-class test set is only for the few-shot experiments.

For each image fed into the 4-block CNN, the last feature map ($64@5 \times 5$) is retrieved for simplex modeling. In order to choose a proper local region for conducting the few-shot experiments, we first test the $N$-way $k$-shot accuracy of different sizes of regions on the validation set. The size of the local regions varies from $2 \times 2$ to $5 \times 5$. [2] The result is shown in Table 1. On the validation set, the models with $2 \times 2$ and $3 \times 3$ regions perform the best on the 1-shot and 5-shot tasks, respectively. We take them to compare with other models on the test set.

Following the same splits proposed by Ravi & Larochelle (2017), we compare the performance of our model on miniImageNet directly with other models. We also conduct the $K$-nearest neighbor on the feature vectors generated from the 4-block model as a comparison (Baseline KNN). The results are shown in Table 2. Using the same pre-trained 4-block CNN, our model performs much better than the baseline KNN, meanwhile outperforms the matching networks and Meta-learner LSTM. However, the prototypical networks are better than our simplex algorithm.

## 4.2 OMNIGLOT

Omniglot dataset for one-shot learning (Lake et al., 2015) contains characters from 50 alphabets ranging from Korean to ancient Greek. Each character is provided with 20 examples handwritten by

---

[2] The $1 \times 1$ scenario does not meet with the definition of a simplex in 5-way 5-shot task, with number of vertices larger than the dimension.

Table 3: Few-shot accuracy on Omniglot.

| Model | 5-way | | 20-way | |
|---|---|---|---|---|
| | 1-shot | 5-shot | 1-shot | 5-shot |
| Baseline KNN | 94.1% | 98.7% | 85.1% | 95.8% |
| Mahalanobis distance | 94.5% | 99.1% | 85.8% | 96.9% |
| MICL (Wright et al., 2008) | 95.4% | 99.1% | 87.3% | 96.9% |
| Siamese networks (Koch et al., 2015) | 97.3% | 98.4% | 88.2% | 97.0% |
| Matching networks (Vinyals et al., 2016) | 98.1% | 98.9% | 93.8% | 98.5% |
| Prototypical networks (Snell et al., 2017) | 97.4% | 99.3% | 96.0% | 98.9% |
| **Simplex (ours)** | 94.6% | 99.1% | 85.7% | 97.0% |

20 online contributors. Omniglot fits the few-shot scenario well: comparing with the large number of categories (1623), the examples (20) are relatively limited, making it difficult to be trained upon with conventional parametric networks.

Following the training setting in Vinyals et al. (2016), we split Omniglot into two parts: 1200 characters for training and the rest for validation and few-shot testing. The embedding 4-layer CNN is almost the same as used for training miniImageNet, except that the output feature map is changed to $64@1 \times 1$ due to the decrease in image size. We compare the $N$-way $k$-shot performance of our model with others. The results are shown in Table 3. Our models are overall comparable to the other state-of-the-art works.

### 4.3 ROBUSTNESS OF SIMPLEX METHOD

Besides, we conduct more experiments on MICL and our model for a further comparison on robustness. MICL has a distortion parameter $\epsilon^2$ in the coding length, i.e. (Wright et al., 2008)

$$L_\epsilon(\mathcal{X}) = \frac{n+d}{2} \log_2 \det \left( I + \frac{d}{\epsilon^2} \Sigma(\mathcal{X}) \right) + \frac{d}{2} \log_2 \left( 1 + \frac{\mu^T \mu}{\epsilon^2} \right), \quad (12)$$

where $\Sigma(\mathcal{X})$ is the corresponding covariance matrix and $\mu$ is the center of $\mathcal{X}$. Through adjusting the parameter, the optimal performance of the model can be obtained on the validation set. We follow the same way as MICL and set up the free parameter in our simplex model. To be specific, let $\lambda_1, \ldots, \lambda_n$ denote the eigenvalues of $\hat{Q}$. It is easy to write $\det(\hat{Q}) = \prod_{i=1}^n \lambda_i$. To include the free parameter, we employ the following expression instead

$$\det \left( I + \frac{d}{\epsilon^2} \hat{Q} \right) = \prod_{i=1}^n \left( 1 + \frac{d}{\epsilon^2} \lambda_i \right). \quad (13)$$

The same computation is also performed for $\det(\hat{P})$.

The experiment is conducted on the different values of distortion $\epsilon^2$. According to the results in Figure 5, it is clear that our model is far more robust than MICL. Although for some value of $\epsilon^2$, the accuracy of MICL is close to our model, the overall performance of MICL is instable with respect to $\epsilon^2$. On the contrary, the performance of our model almost keeps invariant in a wide spectrum of $\epsilon^2$.

### 4.4 ANALYSIS AND DISCUSS

It is worth noting that our simplex metric can also be combined with very recently published works including prototypical networks (Snell et al., 2017) and meta-learning methods, such as (Finn et al., 2017) and (Mishra et al., 2017). For example, the distance measure in prototypical networks can be replaced with our simplex metric and the associated networks can be learned in the end-to-end manner by the supervision of simplex volumes. For few-shot cases, the number of examples in each class is quite limited. So the computational complexity can be well managed. Besides, the

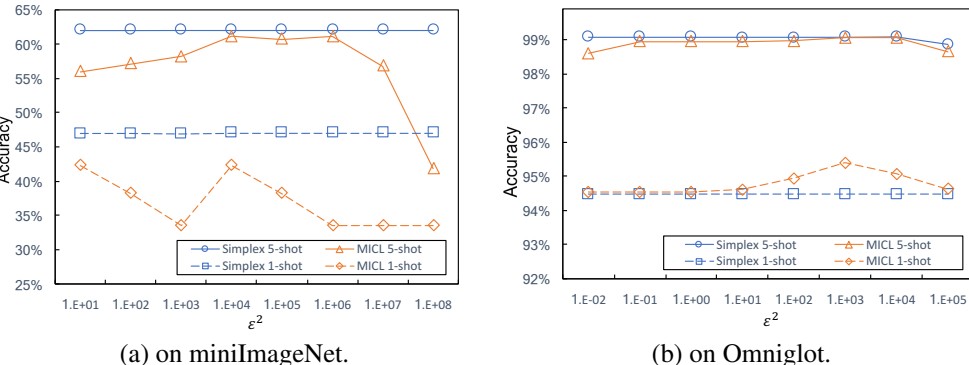

(a) on miniImageNet.  (b) on Omniglot.

Figure 5: Comparison of robustness between our simplex algorithm and MICL with respect to free parameters $\epsilon^2$.

meta learning methods are to learn models between tasks instead of data points. Therefore, they are applicable to improve performance based on the simplex metric. We leave these for further explorations.

On the other hand, MICL (Wright et al., 2008) and our algorithm directly exploit the features of new classes yielded by CNNs that are not retrained, fine-tuned, or performed any relevant refinement of model parameters on new classes. Even so, these two approaches achieve better performance on 5-shot recognition than matching network (Vinyals et al., 2016) and one of meta-learning algorithms (Ravi & Larochelle, 2017) that apply more information of new classes and high-level learning techniques. Both MICL and our simplex algorithm harness the geometric characterization of class structures [3]. Therefore, our work might inspire the interest of exploring geometry to solve few-shot learning problems, which is paid little attention in the field.

## 5 CONCLUSION

In this paper, we designed a novel method to deal with few-shot learning problems. Our idea was from the point of view of high dimensional convex geometry and transformed the learning problem to the study of volumes of simplices. The relation between a test sample and a class was investigated via the volumes of different polytopes. By harnessing the power of simplex, we gave a rigorous mathematical formulation for our approach. We also conduced extensive simulations to validate our method. The results on various datasets showed the accuracy and robustness of the geometry-based method, compared to the state-of-the-art results in the literature.

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
