# OpenReview forum: "Few-Shot Learning with Simplex"
_ICLR.cc/2018/Conference — Reject_

### Official Review · AnonReviewer2 · 2017-11-27
**Interesting approach but missing baselines and state of the art claims wrong**

**Rating:** 4
**Confidence:** 4

**Review:**

This paper proposes a geometric based approach to solving the problem of one-shot and few-shot learning. The basic idea is to use the feature vectors of a particular class to construct a simplex. (I am assuming the dimensions of the vectors are selected so as to exactly construct a simplex? It is not clearly written in the paper). The volume of the simplex is then taken to be a measure of class scatter, and classification happens by assigning the test feature vector to the nearest simplex, where the distances are normalized by the volume of the simplex.

While the approach makes sense, I am not convinced that this geometric method plays an important role in increasing the performance on one-shot/few-shot tasks. In particular, one could try simpler approaches like k-NN where the distances to the cluster centers are also normalized by the variance within the clusters. I would suspect that this method is not superior to this simpler baseline.

The other issue I have with this paper is misleading claims about being state of the art on Omniglot. In particular see Kaiser et al (ICLR 2017), where on 5-way-1-shot an accuracy of 98.4% is reached compared to 94.6% in this paper, and on 5-way-5-shot an accuracy of 99.6% is reached compared to 99.1% in this work. The paper also misses evaluations on various other data sets such as GNMT etc., on which Kaiser et al evaluated their approach.

---

> ### Author Response · Authors · 2017-12-09
> **Thanks for your comment**
>
> According to our experiments on miniImageNet and Omniglot, our algorithm overall outperforms Matching Network (NIPS 2016) and Meta Learner (ICLR 2017) on these two benchmarks, even though our algorithm is based on features produced by a simple CNN and performs no refinement on new classes. This indicates that the simplex idea shows the superiority compared with some of new but more complicated algorithms for few-shot learning.
>
> The algorithm proposed by Kaiser et al (ICLR 2017) indeed outperforms our algorithm on Omniglot. We will make the claim proper in the revision.
> We followed the experimental protocol presented by Ravi & Larochelle (ICLR 2017). So we did not perform experiments on  GNMT. The two benchmarks we used are the datasets that are most frequently used by other researchers for few-shot learning.

---

### Official Review · AnonReviewer3 · 2017-11-29
**Interesting technique but more complicated and lower performance than recent approaches**

**Rating:** 4
**Confidence:** 4

**Review:**

This work proposes a method for few-shot classification that treats a set of embedded points belonging to a class as a simplex. Classification for an unlabeled test point is performed by selecting the class whose augmented simplex has the smallest volume relative to the original class simplex.

Strengths
- The use of simplices for representing classes in few-shot learning is novel.

Weaknesses
- A number of recent related few-shot learning approaches are missing from the related work.
- In light of missing baselines, the proposed method does not perform better than recent few-shot approaches.

I am not an expert on simplices, but the derivations in the paper appear to be correct, with two exceptions: (a) equation 6 appears to be the ratio of C^2(Y U t) / C(Y), (b) equation 6 appears to be missing a minus sign.

The writing of the paper is relatively clear, however there are several important issues:
- Background on metric learning is missing.
- The training loss is not described (i.e. how is the volume ratio from equation 7 converted into a probability distribution over classes?).
- The local feature representation in Section 3 is unclear and should be explained in more detail.
- Related work is missing recent few-shot learning approaches, including MAML [1], Prototypical Networks [2], and TCML [3].

The proposed method is a metric learning approach but it has some additional restrictions relative to other such techniques for few-shot learning such as Matching Networks or Prototypical Networks. One is that the computation of volume ratio involves matrix inversion of P and Q. Another is that the method is not defined when the number of points exceeds the dimensionality of the embedding space. These are not likely to be an issue for few-shot learning, but should be noted as interest in methods that scale gracefully from the few-shot to ordinary classification increases.

Regarding Omniglot results, 20-way 1-shot/5-shot experiments are widely reported in related work but missing from the paper.

When the results are viewed in light of missing baselines, such as Prototypical Networks (68.2% on 5-way 5-shot miniImagenet), the proposed method is more complicated and performs significantly worse.

Overall, the proposed approach is interesting but there are significant issues with both background/related work and performance relative to missing baselines.

[1] Finn, Chelsea, Pieter Abbeel, and Sergey Levine. "Model-Agnostic Meta-Learning for Fast Adaptation of Deep Networks." ICML 2017.
[2] Snell, Jake, Kevin Swersky, and Richard S. Zemel. "Prototypical Networks for Few-shot Learning." NIPS 2017.
[3] Mishra, Nikhil, et al. "Meta-Learning with Temporal Convolutions." arXiv preprint arXiv:1707.03141 (2017).

EDIT: I have read the author's response. The background and related work issues are largely fixed in the latest revision of the paper. Thanks also to the authors for clarifying that training proceeds according to the minimization of cross-entropy loss, rather than a loss based on the simplex. In this case, the novelty of the proposed method then lies in the test-time procedure for making a classification decision when a few-shot episode is encountered. Thus the novelty is relatively low in my opinion. From an experimental perspective, I believe that a comparison of the proposed approach to other test-time classification decision rules is warranted to demonstrate that the simplex rule is better than simpler alternatives (for example, fitting a Gaussian distribution to the support examples of each few-shot class and then assigning a test example to  the class with highest posterior probability). My rating remains unchanged.

---

> ### Author Response · Authors · 2017-12-09
> **Thanks for your comment**
>
> We will cite the papers you listed in the revision. They are excellent works.
>
> But first, I want to make it clear that we did not use simplex volume as the loss to train networks. As Fig. 4 illustrates, we employed a simple 4-block CNN supervised by softmax. For new classes, we only applied the feature maps produced by the last convolution layer, as shown in Fig. 3. Namely, we did not  use any re-training, fine-tuning,  or other manipulations that refine the CNNs for new classes. Even though, our simplex algorithm overall outperforms Matching Network (NIPS 2016) and Meta Learner (ICLR 2017) on two benchmarks.
>
> For Prototypical Networks (PN), our simplex idea can also be applicable according to the formulation of this elegant algorithm, e.g. replacing the distance metric in PN with our simplex metric. Our algorithm can also be combined with Meta-learning algorithms such as MAML and TCML, to improve performance, because they are developed from the different level of learning models for few-shot learning.
>
> About the typos, the squares are missed in equation (3) and a minus sign in equation (6). These will be revised. Thank you for pointing out these typos.
> Besides,  the local feature representation in Section 3 will be presented in more detail.

---

### Official Review · AnonReviewer1 · 2017-11-30
**No Title**

**Rating:** 6
**Confidence:** 4

**Review:**

This paper proposes an approach for few-shot classification based on a geometric idea. The basic assumption is that a query instance will be closest to the polytope corresponding to the correct class than to other classes, where they consider polytopes formed by selecting samples from each class as vertices. As a distance metric, authors consider the variation of the volume of each class-polytope when a query instance is added to the corresponding class. Given that there is not a method to calculate the volume of a general convex polytope, they approximate the polytope by the corresponding simplex convex (convex polytope with the condition of n = d). Fortunately, in the case of the simplex, there is close form solution to obtain the volume.

In general, the paper is well presented and, as far as I know, the proposed idea is novel and sound. Experimentation is correct. Results indicate that the proposed method is able to outperform related state-of-the-art techniques, achieving a reasonable improvement, approx. 1-3% depending of the dataset.

As a drawback, for each query instance, the method needs to estimate the distance to the simplex of each class, therefore it does not scale well with the number of classes. Authors should comment about this issue, in particular, about the computational complexity of the proposed method. Also, in the cases presented in the paper, the selection of the training instances used to calculate the simplex is straight-forward, however, in a more general case, this could be a relevant problem. It will be good to comment about this issue.

---

> ### Author Response · Authors · 2017-12-09
> **Thanks for your comment**
>
> Our algorithm works well for few-shot learning case where sizes of matrices for computing determinants are generally up to 5x5.  The computation complexity is actually low. However,  the cubic complexity caused by determinant will significantly slow down our algorithm if there are much more samples in each class. For such case, our algorithm cannot scale well with the number of classes. But for few-shot learning, this is not an issue.
>
> Actually, using the bordering method of matrix inversion, we can further expand $P$ with $Q$ in formula (7). Theoretically,  we only need to compute the inverse of $Q$ one time for each class during testing.

---

### Decision · Program_Chairs · 2018-01-29
**ICLR 2018 Conference Acceptance Decision**

**Decision:**

Reject

**Comment:**

Reviewers largely acknowledge the novelty of the paper in proposing the use of simplex volume for measuring class scatter in few-shot learning. However there are concerns on missing comparisons with relevant baseline methods, both earlier published work as well as other simpler variants without using the volume (k-NN etc).